# Distribution and Genesis of Organic Carbon Storage on the Northern Shelf of the South China Sea

**DOI:** 10.3390/ijerph191811367

**Published:** 2022-09-09

**Authors:** Liang Chen, Zhengxin Yin, Meng Tang, Tuanjie Li, Dong Xu

**Affiliations:** 1South China Sea Marine Survey and Technology Center, State Oceanic Administration, Guangzhou 510300, China; 2Key Laboratory of Marine Environmental Survey Technology and Application, Ministry of Natural Resources, Guangzhou 510300, China; 3South China Sea Bureau of Ministry of Natural Resources, Guangzhou 510300, China; 4Second Institute of Oceanography, Ministry of Natural Resources, Hangzhou 310012, China

**Keywords:** sediment, organic carbon storage, particle size, the northern South China Sea

## Abstract

The sediments distributed in the marginal seas of the continental shelf are important burial materials for global organic carbon (OC). There have been many estimates of the global continental shelf OC reserves, but due to the limited acquisition of measured data, the estimated results have great uncertainty. The vast continental shelf in the northern part of the South China Sea (SCS) provides a good place for the storage of OC. Based on a large amount of sediment OC data obtained from the northern coast of the SCS, the OC storage in the surface sediment (0~10 cm) in the study area (approximately 8.63 × 10^4^ km^2^) was accurately calculated as 51 Tg. The study area covers different regions, such as estuaries, open seas, strait areas and upwelling development areas, and the OC content of each area is quite different. According to provenance analysis, the source of OC in sediments is mainly from the input of Pearl River runoff. The OC content is significantly higher and less affected by sediment particle size in the Pearl River Estuary and the surrounding areas; meanwhile, the OC content gradually decreases with the distance from the Pearl River Estuary. Far from the western Pearl River Estuary, the sediment OC content is mainly controlled by the particle size of the sediments and is significantly correlated with silt and clay content. The deposition rate is also an important factor affecting the burial of OC, for the high deposition rates correspond to the high levels of OC in the nearshore estuarine areas, as well as the low deposition rate region having low OC content in the sediments even though it has a high productivity of OC, such in as the upwelling sea area on the eastern side of Hainan.

## 1. Introduction

Marine sediment is an important storage site for global organic carbon (OC), and large amounts of OC buried in marine sediments help to reduce the CO_2_ content in seawater, thereby promoting the absorption of more atmospheric CO_2_ by seawater and alleviating the trend of global warming. Although coastal estuaries account for only 0.2% of the global ocean area, their atmospheric CO_2_ fluxes are much higher than those in open sea areas [1]. At the same time, although the continental shelf accounts for only 7–10% of the global ocean area [2], it contributes 10–30% of the primary marine productivity, as well as buries 30–50% of the inorganic carbon and approximately 80% of the OC in the world [3]. After summarizing a large amount of data, Burdige (2007) estimated that approximately 80% of global OC burial occurred on the continental margin where the water depth is less than 2000 m, with a burial flux of 248 TgCyr^−1^ [4]. Additionally, Diesing et al. (2021) summarized previous estimates and concluded that the carbon burial flux on the continental shelf ranged from 45.2 to 300 TgCyr^−1^ [5]. The estimated global carbon storage in marine sediments varies considerably from 87 PgC (top 5 cm) [6] to 147 PgC (top 30 cm) [7] to 3117 PgC (top 1 m) [8]. Due to the high primary productivity and deposition rate, the offshore area has been considered as the richest OC reserve, but the estimation results of different scholars are quite different [5]. Part of the reason for the large difference is the inconsistent selection of methods, such as the inconsistency of the depth [9], but a larger part due to the lack of measured data, and most results have come from model calculations [6,8,10].

At present, a large number of CO_2_ observations at the air–sea interface have been carried out in the northern SCS, which shows that the SCS basin is a weak source of atmospheric CO_2_, and the northern SCS is characterized by a source of atmospheric CO_2_ in the warm season and a sink in the cold season [11,12]. However, under the influence of monsoons and river input, the continental shelf of the northern SCS is a carbon sink as a whole [13]. The northern SCS has complex biogeochemical processes due to the input of Pearl River runoff, and the estimated dissolved organic carbon (DOC) transported to the continental shelf from Lingdingyang in the Pearl River Estuary is 5.3 × 10^8^ gCd^−1^ during the dry season [14]. As an important part of the carbon cycle, sediment carbon burial, has naturally received additional attention. Cao (2017) carried out the detection and analysis of OC and carbon stable isotopes on 32 surface sediment samples collected outside the Pearl River Estuary and eastern Hainan Island, and found that the preservation status of organic matter in surface sediments is different in each region, as well as that organic matter decreases with the increasing specific surface area of the sediment in the deep sea, while the preferential adsorption of fatty acids by clay minerals at some sites is beneficial to the preservation of organic matter [15]. Hu (2006) conducted a study of isotopic composition (d ^13^ C and d ^15^ N), organic carbon (OC) and total nitrogen (TN, organic plus inorganic) on 37 sediment samples from the Pearl River Estuary and the surrounding continental shelf sea, which indicated that the organic material is a mixture from two sources, both terrestrial and marine in this region, but due to the limit of the number of samples, it failed to provide more detailed information of the region [16]. Yang et al. (2022) analyzed the OC characteristics of sediments in a canyon on the continental slope of the northern SCS and found that the OC content in the upper layer of the Yitong Canyon gradually decreased from the upper slope to the middle slope and the lower slope, which were 1.14–2.02%, 1.01–1.46% and 0.60–2.15%, respectively, and the OC content in the upper sediments of the continental slope of the Dongsha Islands is 0.52–1.35% [17]. Therefore, in the northern part of the SCS, most reports have examined the carbon burial mechanism based on representative sites. However, there have been no reports on the storage and distribution of OC in seabed sediments on a large scale in the coastal areas of the northern SCS. Based on OC, grain size, clay mineral and deposition rate data of sediment samples collected from more than 700 sites in the northern SCS, this paper systematically describes the OC storage and its origin in the northern SCS.

## 2. Materials and Methods

### 2.1. Location and Sampling

The South China Sea (SCS), due to its special geographic location, is located between the Qinghai–Tibet Plateau and the western Pacific Warm Pool. There are both land-based inputs from the Qinghai–Tibet Plateau and warm current exchanges from the western Pacific, so the distribution of sediment types is diverse and complex due to the dual influences of terrigenous materials and tropical marine authigenesis. For the study area, it is located on the continental margin; usually, the sources of nearshore sediments are runoff input and coastal weathering, and there are few marine authigenic minerals. The runoff of the Pearl River ranks second in China, with an annual runoff of 3.26 × 10^11^ m^3^ and sediment discharge of 8872 × 10^4^ t [18,19]. Affected by surface circulation, Pearl River-diluted water flows northeastwards in summer, and in winter, it flows in a southwest direction along the coastline driven by the northeast monsoon [20,21]. Chen et al. (2013) analyzed the major elements of sediments in samples consistent with the station positions in this paper and they proposed that the sediment sources in the study area could be divided into three categories [22]. The first category is the Pearl River runoff input material distributed outside the Pearl River estuary and the western Guangdong Sea area, which is the most widely distributed type of sediment in the study area, followed by fan-shaped sediments in the eastern part of the Qiongzhou Strait; the last type comprises sediments with high marine autogenesis outside the eastern part of the Qiongzhou Strait.

Surface sediment samples were collected from 2004 to 2008 in three regions (Figure 1): A—within the Pearl River Estuary, B—outside the Pearl River Estuary and C—in the western Guangdong Sea. The sampling of A and B in the Pearl River Estuary was completed in the spring and summer of 2004 and 2005, and the sampling of C in the western Guangdong Sea was completed in the spring and summer of 2008. The main sampling tools were box samplers and clam samplers. The upper layer (0~10 cm) sediments were collected for testing, and the sampling intervals were mainly 3 km, 5 km and 10 km. A total of 3500 surface samples were obtained, and all samples were tested for grain size. A quarter of the samples (approximately 720) at the surface site (Figure 1) were selected for OC and clay mineral testing. At the same time, some gravity core samples were obtained, and 15 sites were selected to carry out ^210^Pb tests. At the same time, some core samples were tested for volume density and water content.

### 2.2. Test and Analysis

For the sediment particle size analysis, we first removed the organic matter, calcareous cement and biological shell in the sample and then added sodium metaphosphate to disperse the sample by ultrasonic oscillation. The components smaller than 0.063 mm were analyzed by a Mastersizer 2000 laser particle size analyzer, while the coarse particles larger than 0.063 mm were analyzed by the sieving method. The mean particle size was calculated by using the formula of the moment method, then the particle size was calculated by using the standard of the φ value; finally, the sediment was divided into sand (0–4 φ), silt (4–8 φ) and clay (>9 φ) according to the particle size [24].

According to the Chinese national standard GB17378.5-2007 18.1 [25], the determination of OC in the Pearl River Estuary sea area (A and B) adopted the ferrous ammonium sulfate volumetric method. A certain amount of potassium dichromate–sulfuric acid solution was used to oxidize OC in the sample and then was titrated by ammonium ferrous sulfate standard titration solution to calculate the OC content. The sedimentary OC around the western Guangdong Sea (C) was detected by the oxidative pyrolysis–potentiometric method according to the Chinese national standard GB17378.5-2007 18.2 [25]. Under oxygen-enriched conditions, the sample was decomposed and burned at 1000 °C. Carbon and organic matter in carbonates generated carbon dioxide, which was imported into the sodium hydroxide solution absorption cell for absorption, and the potential change of the absorption cell solution was measured at the same time. Then, the total carbon content in the sample was calculated according to the calibration curve. The carbonate in the sample was decomposed with dilute hydrochloric acid, and the carbon content in the sample was determined after the inorganic carbon was removed so that the OC in the sample could be determined again [26]. According to quality control, for the results obtained by the two methods, when the mass fraction of OC was greater than 5%, the error was no more than 0.5%; when the mass fraction was less than 1%, the error was no more than 0.3%; when the mass fraction was between 1% and 5%, the error was no more than 0.4%. Therefore, the two methods have good reliability and comparability.

Clay minerals were analyzed by X-ray diffraction according to China’s oil and gas industry standard SY/T 5163-2010 [27]. A naturally oriented sheet (N sheet) was prepared after the separation and purification of clay minerals, and an ethylene glycol-saturated sheet (EG sheet) was prepared after X-ray scanning. After the second X-ray scanning, it was heated in a muffle furnace to 550 °C and maintained at a constant temperature for 2 h, then cooled to room temperature naturally to make a heating sheet (T sheet), and the X-ray scanning was performed again. After the analysis of the graph, the clay content (illite, smectite, kaolinite and chlorite) was obtained, and the amount of clay in the whole sample was obtained by the ratio of clay.

The leaching method was used for the chemical analysis of ^210^Pb. After the sample was prepared, it was placed in a detector equipped with a gold–silicon surface barrier α probe for measurement, and the α spectrum was analyzed by a multichannel energy spectrometer to determine the radioactivity ratio of ^210^Po. Then, according to the hypothesis of a radioactivity balance between ^210^Pb and daughter ^210^Po, the ^210^Pb radioactivity ratio was further calculated to obtain the deposition rate [28].

Some core samples were tested for volume density [29], which was carried out by the ring knife method according to the Chinese national standard GB/T50123-2019 [30]. The sample was filled into the ring knife and weighed. The wet density of the sample was obtained after dividing the volume, and the water content of the sample was obtained by drying the sample.

It is generally believed that fine-grained sediment has a high protective effect on organic matter due to its high specific surface area [31,32]. Therefore, after obtaining all kinds of data, the content and composition of OC and fine particles were analyzed emphatically. These analyses included contour maps of mean particle size, clay mineral content, illite, kaolinite, chlorite and OC content, covering the sample collection area. Data points of mean particle size, clay mineral, illite, kaolinite, chlorite and OC contents have been interpolated to build layered diagrams based on the kriging interpolation method, with a cell size of 1.1 km × 1.1km.

The calculation of carbon storage uses the following formula [9]:TC stock (g) = Σi C stock (g/m^2^) i × Area (m^2^) i
C stock (g/m^2^) = [TOC] (%) × BD (g/cm^3^) × depth (cm) × 10^2^
where C stock (kg/m^2^) is the OC storage per unit area, [TOC] (%) is OC content, BD (g/cm^−3^) is dry sample volume density, and depth (cm) is calculation depth. The calculation depth for the surface sampling depth is 10 cm. TC stock (kg) is the total OC stock in the study area. Area represents the spread of OC reserves in different regions.

At last, the correlation the analysis of sedimentary OC with the mean particle size (Mz), sand content, silt content and clay content was carried out in three areas within the Pearl River Estuary (A), outside the Pearl River Estuary (B) and western Guangdong waters (C) (Figure 1); the correlation analysis of OC content with illite, smectite, kaolinite and chlorite was also carried out by the same regions except the smectite in area C.

## 3. Test Results

### 3.1. Characteristics of Grain Size for Sediments

There are various types of sediments distributed from clay to sand in the research area, and the mean particle size of sediments is mostly between 3 and ~8 φ. In the Pearl River Estuary (A), the sediments are dominated by fine silt with 7~8 φ of mean particle size. However, there is a patchy high-value area outside the Pearl River Estuary (B), and the sediments are mainly fine silt with a particle size between 3 and 4 φ. The large area of the western Guangdong Sea is dominated by silt with a particle size of 7~8 φ, while the sediments in the Qiongzhou Strait and the eastern part are the coarsest in particle size with a particle size less than 3 φ and are mainly composed of coarse and medium sand (Figure 2).

### 3.2. Distribution of Clay Minerals

The fine-grained components in the sediments are mainly composed of clay minerals (Figure 3). According to the mapping graphic analysis of the obtained kaolinite, chlorite and illite contents, the distribution of kaolinite shows an obvious trend of decreasing from the nearshore to the sea, and the high values are mainly distributed in the Pearl River Estuary and the west coast. The distribution of chlorite does not show an obvious regularity, and the outer sea has a slightly larger distribution than the nearshore overall, while the distribution of illite is obviously larger in the eastern area than in the western area.

### 3.3. Distribution and Storage Estimation of OC in Sediments

The analysis results show that the content of OC in surface sediments in the northern SCS is mostly between 0.2% and 1% (Figure 4). The high value area is mainly located near the Pearl River Estuary, and most other areas are below 1%. The low-value areas are mainly located in the Qiongzhou Strait and the eastern side of Hainan Island.

The wet density ranges from 1.50 to 1.77 g/cm^−3^ with an average of 1.64 g/cm^−3^. The average water content is 64.65%; therefore, the average BD is 1.03 g/cm^−3^.

The calculation methods of sediment areas with different OC contents are as follows. The OC content of sediments in the study area is mainly between 0 and 1.8%. First, the OC content was interpolated and then plotted at equal intervals of 0.2%. The area of each plot with OC contents of 0–0.2%, 0.2–0.4%, 0.4–0.6%, 0.6–0.8%, 0.8–1.0%, 1.0–1.2%, 1.2–1.4%, 1.4–1.6% and 1.6–1.83% was calculated (Table 1).

According to the formula, the OC storage in the 10 cm surface layer in the study area is about 5.1 × 10^13^ g, namely, 51 Tg. The area of the global continental shelf is 26 × 10^6^ km^2^ [33]. According to the data in this paper, the global continental shelf carbon storage is 1533 × 10^13^ g, namely, approximately 15 PgC. This is more consistent with LaRowe’s estimate, and he proposed that the 0–10 cm OC storage in the global shelf sediments is 9.6–25 PgC [34], which is also close to the 18.5 PgC estimated by Duarte et al. (2005) [33].

A total of 15 stations were tested for their deposition rates (Figure 5), which showed the highest value is 4.38 cm/a near Humen in the Pearl River Estuary, the lowest value is 0.12 cm/a on the eastern side of Hainan Island, and the average value is 1.06 cm/a. The sedimentation rate in the survey area is characterized by a decreasing trend from the inner Pearl River Estuary to the offshore area; also, the deposition rate in coastal area is larger than that in the offshore area in the eastern and western of Pearl River Estuary, and there is a slightly high deposition rate area in the eastern Qiongzhou Strait. Meanwhile, the lowest deposition rate is located on the eastern side of Hainan Island.

## 4. Discussion

### 4.1. OC Sources

The sources of OC in nearshore areas mainly include terrestrial, offshore and authigenic sources, and estuarine areas are generally dominated by terrestrial sources; especially for areas with broad continental shelves and abundant runoff, the organic matter in the waters and sediments of continental shelf areas is generally dominated by sources from rivers and estuaries [35]. However, the OC produced by marine biological action in some estuarine areas can exceed the input from rivers or open seas [36,37]. With increasing distance from the estuary, marine and autogenic organic matter gradually increased; at the same time, a large number of mesoscale vortices developed in the northern South China Sea, which contributed to the formation of organic carbon and can also bring abundant marine OC to the study area [38]. Judging from the gradient change in OC content in the study area, the OC content of sediments is closely related to the input of terrigenous materials in the Pearl River Estuary and is mainly controlled by the input and distribution of terrigenous materials. However, in the areas on the eastern side of Hainan Island, the Pearl River flow can hardly affect this area, so the sediments are dominated by marine sources and land-based materials from Hainan Island.

### 4.2. Factor Analysis of Carbon Storage

OC in seabed sediments mainly comes from the deposition of organic matter in water; however, OC in water is degraded with different degrees in the process of sedimentation due to various factors, such as its being decomposed into smaller organics and further degraded to dissolved inorganic carbon [39]. OC entering sediments may also be remineralized and removed in the early diagenetic process [40]. Finally, OC stored in marine sediments accounts for only 0.5% of the production totally on the earth [4,31]. Nearshore sediments are also susceptible to resuspension by various disturbances, such as typhoons, trawling and other human activities that resuspend OC into seawater [41]. The seawater–sediment interface is generally from a few centimeters to decimeters [42]; a series of physical, chemical and biological processes at the interface will have various effects on the burial of OC, such as the reworking of sediments [43], or being aerobically (i.e., with oxygen) or anaerobically digested (with alternative electron acceptors such as nitrate, metal oxides and sulfate) [44,45]. In addition to reducing the degradation of OC, the excavation and downwards migration of some benthic organisms in the sediments will also increase the content of organic matter at the bottom [46]. Therefore, the burial and preservation of organic matter in seabed surface sediments is extremely complex, especially at continental shelf edges with strong land–sea interactions.

#### 4.2.1. The Effect of Sediment Particle Size on OC

The particle size usually considered as an indicator for the sedimentary hydrodynamic strength and has a dominant control on sediment elements in the coastal area. Continental shelf sediments are mainly composed of terrigenous clasts and clays; except for the black carbon layer, with extremely high content in a few layers [47], most of the time OC is generally stored in the sediments as minor components, and therefore, the preservation of OC is largely affected by the type of sediments. The type of sediment is mainly determined by the particle size, which reflects the source of the material and, especially, the hydrodynamic environment in which deposition occurs [48]. When the hydrodynamic force is strong, the sediment is mainly composed of coarse-grained components, and OC in seawater is mainly composed of dissolved OC and particulate OC; comparing the density differences between organic matter (~1 g cm^−3^) [49] and mineral grains (≥2.5 g cm^−3^) [50], it is difficult for OC to settle and be buried under strong hydrodynamic forces.

The particle size has an important influence on the OC content (Figure 6); the Mz value (φ) of sediments in the survey area is positively correlated with the OC content, and the sand content is negatively correlated with the OC content, while the silt and clay are positively correlated with the OC content. From the perspective of regional distribution, the correlation outside the Pearl River estuary is lowest, followed by that inside the Pearl River Estuary due to the high overall OC content, which reflects less control by particle size or hydrodynamics. In the areas where the OC is not too rich in the western Guangdong Sea and northeastern Hainan waters (C), the correlation is significantly higher, indicating that the size of sediment particles basically determines the content of OC in this region.

#### 4.2.2. Effects of Clay Minerals on OC

The particle size analysis results show that fine-grained components in sediments are more likely to coexist with OC. Although the mechanisms of OC binding to mineral surfaces are currently unclear, various adsorption mechanisms (van der Waals interactions, ligand exchange and cation bridging) should play an important role [51]. Fine components in sediments are mainly clay minerals, and the correlation analysis of OC content with illite, smectite, kaolinite and chlorite by region shows that in the survey area, except for smectite outside the Pearl River Estuary, all other clay minerals are positively correlated with OC. Among all types of clay minerals, kaolinite has the strongest correlation with OC, followed by illite and chlorite, and smectite has the weakest correlation (Figure 7).

Through experimental analysis, Satterberg et al. (2003) proposed that montmorillonite (a member of the smectite family) is easier to combine with OC than chlorite and kaolinite [52]. However, according to a study on the relationship between clay minerals and OC in the northern SCS [53], land-derived OC can be stripped from the surface of montmorillonite by emission, diffusion and deposition in the open sea, which is consistent with the decrease in the correlation between smectite and OC from the inner to outer mouth of the Pearl River Estuary. To a certain extent, this also shows that the distribution of OC around Pearl River Estuary is partially controlled by land-derived materials imported from the Pearl River Estuary.

#### 4.2.3. The Effect of the Deposition Rate on OC

The deposition rate is another important factor affecting the burial of OC, and it is generally suggested that a higher deposition rate more easily preserves organic matter [54,55]. The deposition rate in the study area is obviously higher in the estuary than in other areas, and the deposition rate in the top of Pearl River Estuary is the highest. From the Pearl River Estuary to the western sea area, the deposition rate shows a decreasing trend, and the OC content also decreases, and the same trend is also shown from the Pearl River Estuary to the open sea. On the whole, the study area is also consistent with the characteristics of high OC content with higher deposition rates (Figure 8).

The lowest deposition rate area in this study is on the eastern side of Hainan Island, where upwelling develops from April to September each year [56]. Since coastal upwelling areas are known to contain more CO_2_, when upwelling develops, a large amount of deep CO_2_ will be carried to the surface layer and released into the atmosphere [57]. At the same time, the bottom water also carries a large amount of nutrients [58], which can promote the development of surface productivity. When the biological action is strong, it will reduce the concentration of CO_2_ in the surface water [57]. Although it is difficult to determine the source or sink of atmospheric CO_2_ by upwards flow, it is undeniable that upwelling promotes surface primary productivity. Therefore, there should be abundant OC sources in this area, but the content of OC in sediments is extremely low. On the one hand, when upwards flow develops, there is a high flow rate at the bottom, and a large amount of organic matter is removed. On the other hand, sea-source OC is more easily degraded than terrestrial OC [59]. Therefore, the deposition rate and OC content are one of the lowest in the study area.

## 5. Conclusions

Although only a small part of the organic matter in ocean water can eventually be preserved in seabed sediments, it is still an important burial site in the global carbon cycle and an important way for CO_2_ to be removed from the biosphere due to the large area of the ocean. Analyzing a large number of sediment samples obtained from the continental shelf in the northern SCS, we summarized the following conclusions:(1)The OC storage was accurately calculated to be 51 Tg in a sea area of approximately 8.63 × 10^4^ km^2^ for the top 10 cm of sediments.(2)The source of OC in sediments in the study area was mainly input from the Pearl River. The content of OC in sediments is much higher in the Pearl River Estuary and the surrounding areas, and it is not strongly affected by sediment types. The sediment types in the western part of the study area have an important impact on the distribution of OC.(3)The correlation between smectite and OC decreased from the Pearl River Estuary to the offshore area, indicating that the OC from the land source on the smectite surface was exfoliated away easily.(4)The deposition rate is another important factor affecting the OC content in this area, for the higher OC content with a high deposition rate in the sea area.(5)Although the deposits of OC in the northern shelf of the SCS were obtained in this study, there are great discrepancies for the distribution and carbon storage mechanisms of OC in different areas, so the estimation of OC in a small area often results in large errors especially in the vast SCS; therefore, more accurate field investigations should be proposed to obtain the reserves of OC over larger areas.

## Figures and Tables

**Figure 1 ijerph-19-11367-f001:**
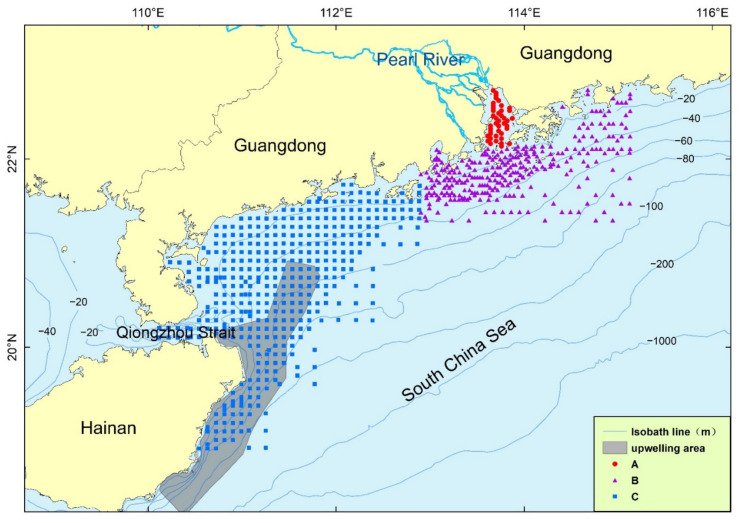
Study area and station distribution. A: Sampling station in the Pearl River Estuary. B: Sampling station outside the Pearl River Estuary. C: Sampling station around the western Guangdong Sea. Upwelling area modified from [23].

**Figure 2 ijerph-19-11367-f002:**
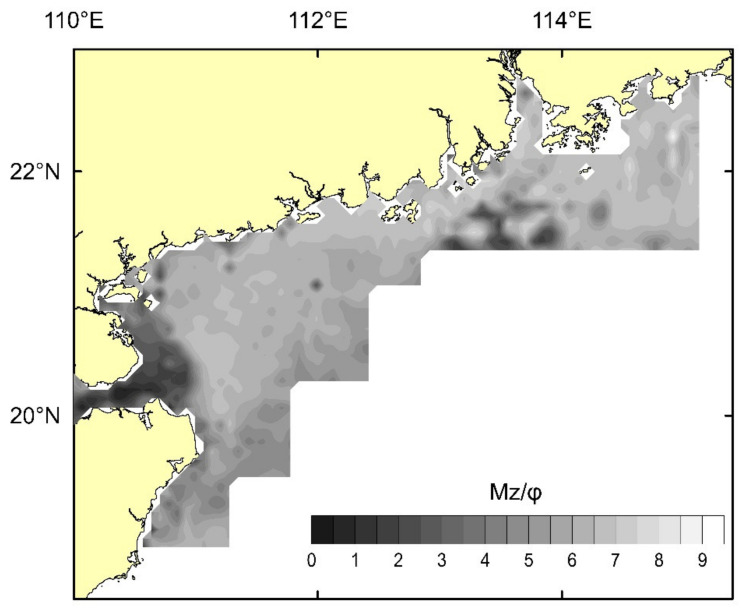
Mean particle (Mz) size distribution of the sediments.

**Figure 3 ijerph-19-11367-f003:**
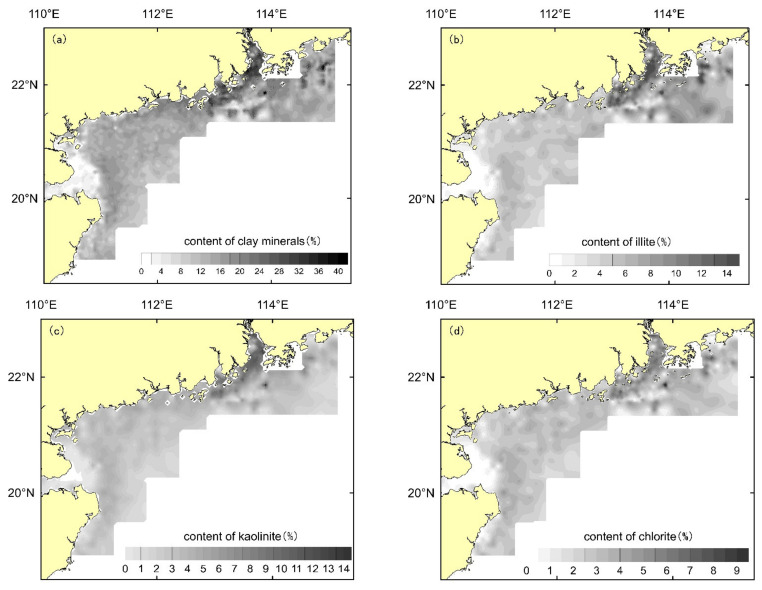
Distribution of clay minerals. (**a**), distribution of clay minerals, (**b**), distribution of illite; (**c**), distribution of kaolinite; and (**d**), distribution of chlorite.

**Figure 4 ijerph-19-11367-f004:**
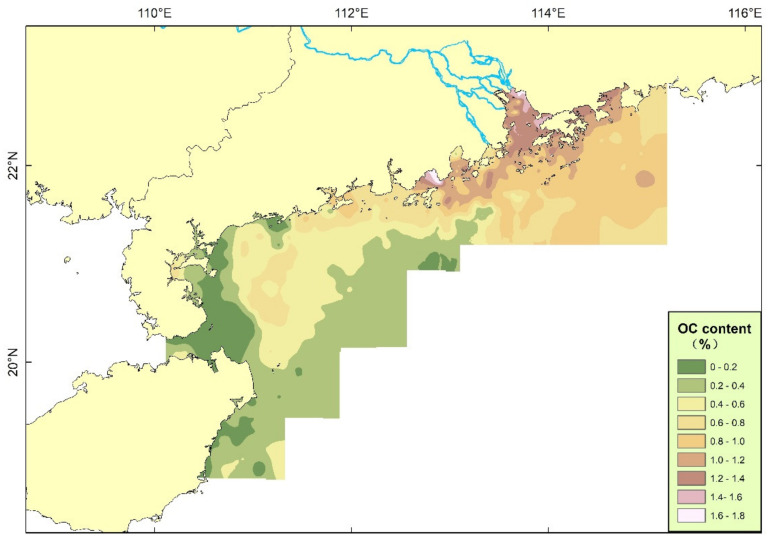
Distribution of OC.

**Figure 5 ijerph-19-11367-f005:**
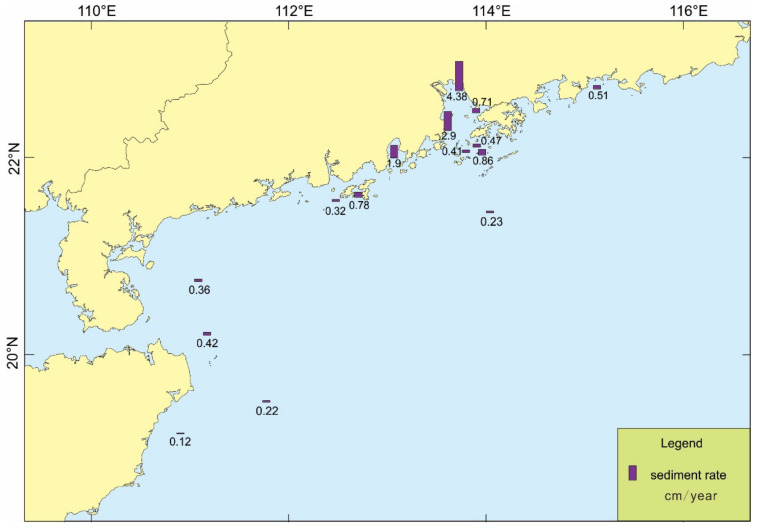
Distribution of sediment rates in the study area.

**Figure 6 ijerph-19-11367-f006:**
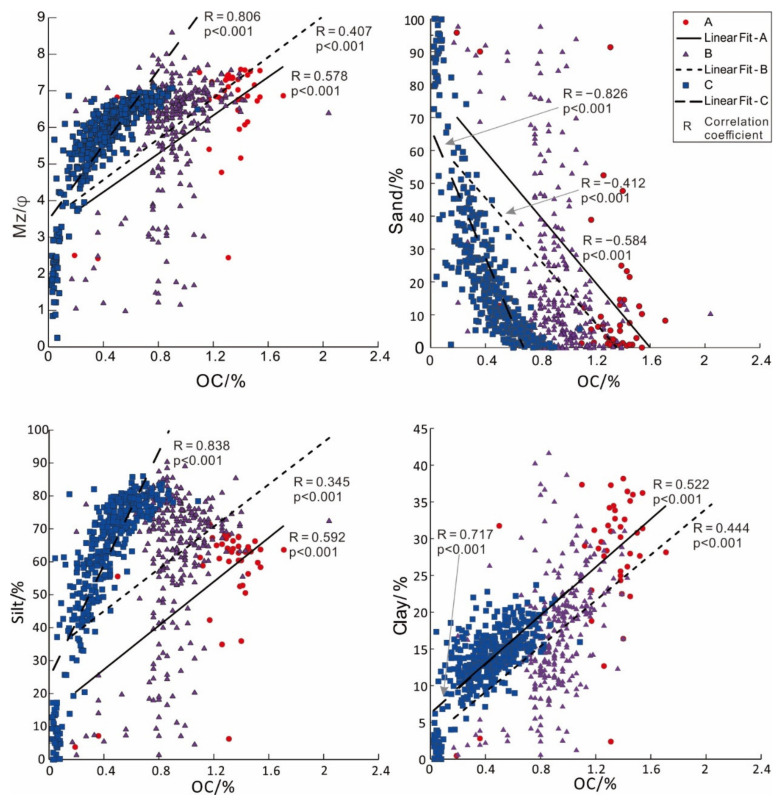
Correlation analysis between OC and particle size in different sea areas.

**Figure 7 ijerph-19-11367-f007:**
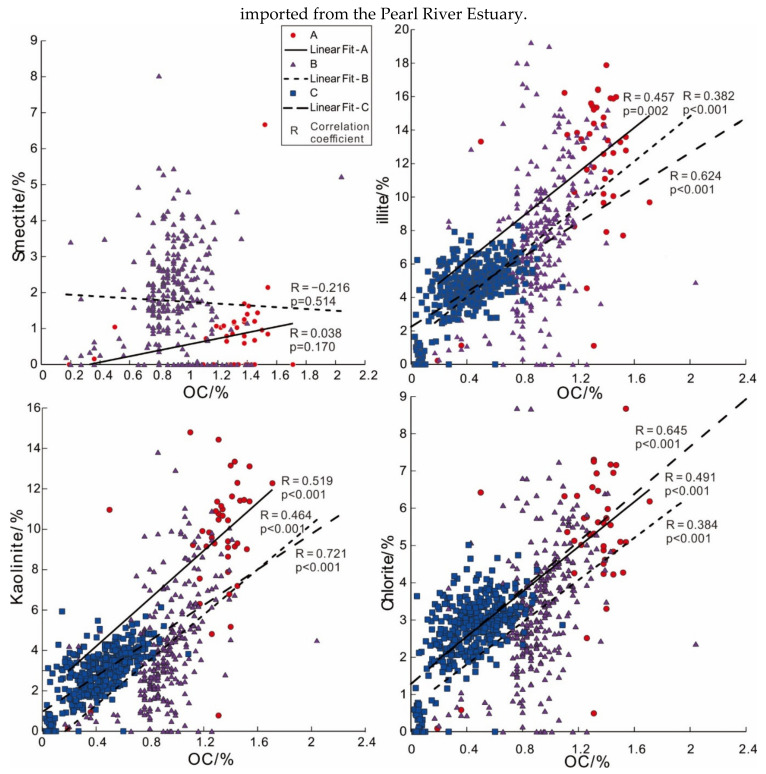
Correlation analysis of OC and clay minerals in different sea areas.

**Figure 8 ijerph-19-11367-f008:**
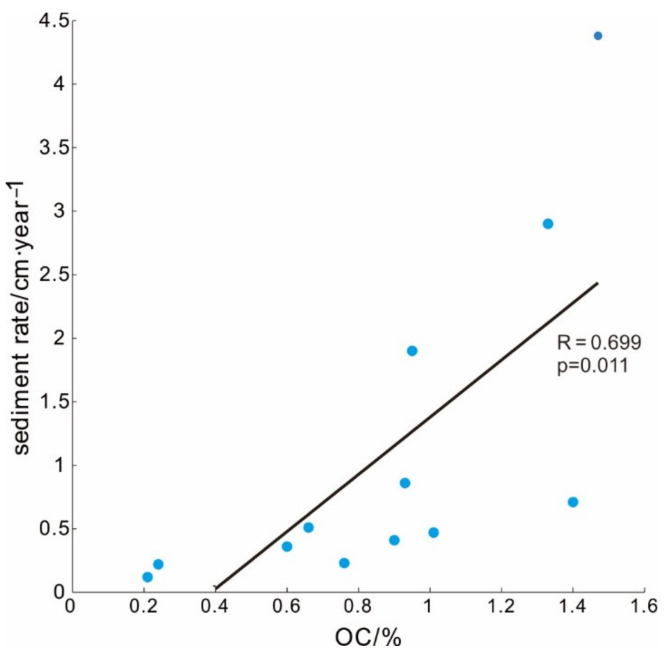
Correlation analysis of OC and sediment rate in different sites.

**Table 1 ijerph-19-11367-t001:** Calculation of organic carbon storage.

NO.	Area/ km^2^	OC /%	BD /g·cm^−3^	Depth/cm	Clay Content/%	Illite /%	Kaolinite /%	Chlorite /%	Smectite /%	Deposit Rate /cm·Years^−1^	C Stock /Tg
1	8221.18	0–0.2	1.03	10	5.74	1.76	1.18	1.00	0.90	/	0.85
2	24882.45	0.2–0.4	1.03	10	12.19	4.42	2.36	2.52	0.44	0.17	7.69
3	15318.94	0.4–0.6	1.03	10	14.52	4.90	3.13	2.95	1.35	/	7.89
4	14382.27	0.6–0.8	1.03	10	15.69	5.91	3.53	2.99	1.80	0.37	10.37
5	16105.90	0.8–1.0	1.03	10	17.41	7.25	3.94	3.26	2.27	1.06	14.93
6	4592.51	1.0–1.2	1.03	10	21.38	9.43	6.01	4.15	1.37	0.47	5.20
7	2390.49	1.2–1.4	1.03	10	26.03	11.77	8.33	4.84	0.63	1.81	3.20
8	360.48	1.4–1.6	1.03	10	30.64	13.16	10.42	5.87	1.20	4.38	0.56
9	121.20	>1.6	1.03	10	22.55	7.29	8.38	4.27	2.61	/	0.21
SUM											50.90

## Data Availability

There are many types of data, and the quantity is large and complex in this study. Therefore, they are available on request from the corresponding author.

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
