# Peer review of "Distribution and Genesis of Organic Carbon Storage on the Northern Shelf of the South China Sea"

_ijerph, 2022, doi:10.3390/ijerph191811367_

Round 1

Reviewer 1 Report

General comment:

The manuscript from Chen et al. ‘Distribution and genesis of organic carbon storage on the northern shelf of the South China Sea’ is an interesting manuscript dealing with a topic which has gained momentum in the last 20 years, carbon storage. The aim is clearly set in the introduction, to describe the origin of organic carbon (OC) storage and its origin in the northern South China Sea (SCS). Despite these facts, the manuscript requires great improvement before being considered for publication, as the introduction requires a bit of polishing, and the methods and results sections are incomplete, as part of the information which should be there is found in the discussion. As for the discussion, it feels as a continuation of the results, as an interpretation of what was found through the analysis the authors did. As such, this section is the one which will require the most work.

The introduction is quite short and concise, divided into two paragraphs. While I usually think it is good to opt for this strategy, the authors are missing a bit more context and a clearer connection between the topic of marine sediments being an important storage of OC, how SCS fits within this context and its role within it.

The methods are explanatory, but are missing references for some of the analysis used, I also had my problems, to see that data presented in the results section, also included description of methods not accounted for in the corresponding section, or the use of terms which should have also been better introduced in the methods section.

As for the results, part of the data presented, wasn’t included in the methods, e.g., clay mineral types, precedence of the sedimented material, calculations of OC, parameters to build Fig. 2-4. Additionally, there are bits of discussion which clearly do not correspond to the results section.

In the discussion section, new figures and analyses are included, which should have been previously explained. Furthermore, the results of these analyses which are used to partly support the conclusions, are not properly or completely presented. Additionally, the discussion appears more as a description of new results, rather than an interpretation of facts or a proper discussion of the authors findings in context with those of other studies.

Based on the above-mentioned general comments, and the list of specific comments (found further below), my overall recommendation is that this manuscript is to be reconsider after major revision.

Specific comments:

-        Comment #1, Introduction: As it is now, the introduction touches 2 topics: 1) the carbon burial in sediments, and how these have been estimated, and; 2) OC variability in the SCS. A clear connection between both topics is clearly missing, which makes the reader wonder, what is the importance of the SCS in the context of climate change? Why is it important to investigate the OC stocks and precedence?

-        Comment #2, Line 36: You should make use of the acronym for organic carbon, and use it consistently throughout the manuscript.

-        Comment #3, Line 55: The phrase ‘In the context of global climate change’ is out of place when the following sentence only describes the SCS. My recommendation is to couple the out-of-place phrase with the fact that the SCS works as both carbon sink and source. The description of the study area can be moved to the methods section.

-        Comment #4, Methods: While some methods are relatively known for non-experts, other are not. For the later, I think that it would be sensible to provide references, e.g., for the oxidative pyrolysis-potentiometric method, the leaching method, and the ring knife method.

-        Comment #5, Line 111-127: Why did you adopted two methods for measuring OC? Are these comparable? It’s necessary to clarify this, as possible differences in terms of OC concentrations between areas/samples might be due to a discrepancy in the methods, rather than discrepancies due to different environmental settings.

-        Comment #6: Wouldn’t have it made sense to test for gradients in either relation to distance from the coast, or from riverine input? Something which is rather mentioned in the subsequent sections, but not formally analyzed.

-        Comment #7: Why not test for differences between locations, this could potentially help to improve the extrapolation of OC stocks for other coastal areas. Also, to clearly show, e.g., difference between river-influenced regions and ‘oceanic’ regions.

-        Comment #8, Line 128: It would be sensible to add the clay minerals to be analyzed (three of them are introduced in the results, and one in the discussion), as well as to point out how each X-ray scanning contributes co calculating clay content and that of each mineral.

-        Comment #9, Line 148-156: Most of this is discussion rather than results. The only arguable result, is the precedence of material, but there is no mention of a method used to account for this. This being part of the main aim of the manuscript, should be already introduced in the methods and better shown in the results.

-        Comment #10, Fig. 2-4: If you have point data, how did you build these maps? The characteristics of the maps (e.g. extension, cell size, interpolation method, cell construction) needs to be mentioned in the methods section.

-        Comment #11, Line 168: This is the first mention of the mapping graphic analysis, which should have been already introduced in the methods section.

-        Comment #12, Line 180: According to the contents of Line 195-199, the OC contents of Fig. 4 were calculated (the method not described in the corresponding section). Furthermore, it appears that at least one third of the shelf has OC contents >1%, contradicting this statement.

I suggest you mention the whole range and, rather than referring to Fig. 4, either prepare a new map or a table sowing the OC ranges of each region (for the table also a spread measure). In case you opted for a table, you could also include absolute values for clay content, clay mineral content, sedimentation rate.

-        Comment #13, Line 203-205: This should be part of the discussion section.

-        Comment #14, Section 4.1: The section provides a mix of published data and what appears to e own data. The later are not properly presented in the results nor included in the methods. That this is presented in both sections is key to properly discuss your observations with those of previous studies.

In case I’m mistaken, and all of what’s here included comes purely from literature, there is still information or a connection to your description of OC inventories missing, as this was the aim of your manuscript.

-        Comment #15, Fig. 6: The circulation you show here, I assume is the summer circulation, correct? This needs to be clarified. Furthermore, if you show this for summer, you should do the same for winter. I recommend having this as supplementary material rather than a figure for the main manuscript, as it is rather secondary and only used to provide a visualization of a short statement which is not further considered in the discussion.

-        Comment #16, Line 224: Remove ‘once’ from ‘Chen et al. () once analysed’.

-        Comment #17, Line 225: Replace ‘believed’ with ‘proposed’.

-        Comment #18, Line 250: There are examples (1-3) and a reference (at least) missing for the statement ‘OC in water is degraded to different degrees due to various factors’.

-        Comment #19, Line 251-252: The percentage of the production represented as OC stored in marine sediments varies regionally, this should be clearly mentioned.

-        Comment #20, Line 255: Reference on depth of sediment-seawater interface is missing.

-        Comment #21, Line 256: Examples of physical, chemical and biological processes affecting OC burial are missing. Also, a pertinent reference is missing.

-        Comment #22, Line 266-267: While this connection can be inferred from connecting Fig. 2-5, no formal analysis supporting this is presented in the methods or results section.

-        Comment #23, Line 267-268: The sentence is a bit complicated. I recommend editing it, e.g.: Sediment particle size reflects the source of the material and, especially, the hydrodynamic environment in which deposition occurs.

-        Comment #24, Line 270-271: Which one of the two dominate? This sentence needs to be clearer. The density of OC where? How do you know, when no values have been properly provided?

-        Comment #25, Line 273-284: This is a whole new part of the study, which wasn’t included anywhere else. The correlation analysis must be included in both methods and results sections.

This is also the first time that concentration of sand and silt are mentioned, why were these not better described in the results?

Another thing to consider, especially if you mention significant differences, is the use of statistics to prove these differences. As for the correlations, I would also have expected a measure of significance. It wouldn’t be surprising that correlations with an R-value between -0.6 and 0.6 were not significant, especially if the analysis was a bit more robust by including permutations or adjusting the p-values. If this were the case, any conclusions reached would need to be revised and reformulated accounting for this.

-        Comment #26, Line 290: Various mechanisms, such as?

-        Comment #27, Line 291-296: Same comment regarding the correlations as comment #25. Furthermore, I think this is the first time ‘montmorillonite’ is mentioned. If this was part of the clay analysis, it should be included in the methods and results section.

-        Comment #28, Line 297: Believed, found, or proposed?

-        Comment #29, Line 303-304: Another mention of precedence without before providing data or references on the topic.

-        Comment #30, Line 309: Believed, thought, or suggested/proposed? A reference supporting the statement is missing.

-        Comment #31, Line 313-315: This could also be supported by correlating e.g. values of OC and deposition (considering stations for which both are available), or correlating OC carbon with distance from the coast or estuary.

-        Comment #32, Line 317-319: There should be literature supporting one or both statements, as such, a reference (at least) should be included.

-        Comment #33, Line 336-337: how is it that this values for area and stock differ from those presented in the results?

Reviewer 2 Report

Authors made an attempt to quantify the genesis and distribution of organic carbon the northern shelf of SCS. This study important to understand the dynamics of estuary sediment transport and its interaction with adjacent upwelling zones. 

1) Authors need to discuss about the seasonal variation of organic carbon in this region, upwelling zone does show seasonal variation.

2) How sediment rates are estimated, need to provide detailed methodology?

3) What is role of mesoscale eddies, how it influences the sediment transport in this region.

4) There are similar studies conducted in the region (references given below) need to refer them and how present study results are different from the earlier studies.

Zhang, M., Wu, Y., Wang, F., Xu, D., Liu, S., & Zhou, M. (2020). Hotspot of organic carbon export driven by mesoscale eddies in the slope region of the northern south China sea. Frontiers in Marine Science7, 444.

Hu, J., Jia, G., Mai, B., & Zhang, G. (2006). Distribution and sources of organic carbon, nitrogen and their isotopes in sediments of the subtropical Pearl River estuary and adjacent shelf, Southern China. Marine chemistry98(2-4), 274-285.

Round 2

Reviewer 1 Report

The authors have put a great deal of work on improving the manuscript, which clearly shows in their second version. From my side, the main issues I saw with their manuscript have been tackled and improved.

One last minor issue to fix/adapt would be the units provided in Table 1 to fit the rest of the literature and make easier read the numbers, by avoid having to resort to using, e.g., 106 or E+11. In that regard, area could be changed to km2 or ha whereas C Stock could be provided in the more common Tg km-2, Mg km-2, or Mg ha-1.

Author Response

   Thank you very much to your comments and suggestions. The suggestions are very helpful for the manuscript. So, it has been carefully revised.

 Comment: one last minor issue to fix/adapt would be the units provided in Table 1 to fit the rest of the literature and make easier read the numbers, by avoid having to resort to using, e.g., 106 or E+11. In that regard, area could be changed to kmor ha whereas C Stock could be provided in the more common Tg km-2, Mg km-2, or Mg ha-1.

Response: We have modified inappropriate expressions and have adopted commonly used expressions. Thanks a lot.